

# Measuring multi-year changes in the Symbiodiniaceae algae in Caribbean corals on coral-depleted reefs

Ross Cunning[1], Elizabeth A. Lenz[2] and Peter J. Edmunds[3]

[1] Conservation Research Department, John G. Shedd Aquarium, Chicago, Illinois, United States
[2] University of Hawai Sea Grant College Program, University of Hawai at Mānoa, Honolulu, Hawaii, United States
[3] Department of Biology, California State University, Northridge, Northridge, California, United States

Corresponding author
Ross Cunning,
ross.cunning@gmail.com

## ABSTRACT

Monitoring coral cover can describe the ecology of reef degradation, but rarely can it reveal the proximal mechanisms of change, or achieve its full potential in informing conservation actions. Describing temporal variation in Symbiodiniaceae within corals can help address these limitations, but this is rarely a research priority. Here, we augmented an ecological time series of the coral reefs of St. John, US Virgin Islands, by describing the genetic complement of symbiotic algae in common corals. Seventy-five corals from nine species were marked and sampled in 2017. Of these colonies, 41% were sampled in 2018, and 72% in 2019; 28% could not be found and were assumed to have died. Symbiodiniaceae ITS2 sequencing identified 525 distinct sequences (comprising 42 ITS2 type profiles), and symbiont diversity differed among host species and individuals, but was in most cases preserved within hosts over 3 yrs that were marked by physical disturbances from major hurricanes (2017) and the regional onset of stony coral tissue loss disease (2019). While changes in symbiont communities were slight and stochastic over time within colonies, variation in the dominant symbionts among colonies was observed for all host species. Together, these results indicate that declining host abundances could lead to the loss of rare algal lineages that are found in a low proportion of few coral colonies left on many reefs, especially if coral declines are symbiont-specific. These findings highlight the importance of identifying Symbiodiniaceae as part of a time series of coral communities to support holistic conservation planning. Repeated sampling of tagged corals is unlikely to be viable for this purpose, because many Caribbean corals are dying before they can be sampled multiple times. Instead, random sampling of large numbers of corals may be more effective in capturing the diversity and temporal dynamics of Symbiodiniaceae metacommunities in reef corals.

# INTRODUCTION

Over the last several decades, symbiotic reef corals have been dying at high rates in tropical seas (*Hughes et al., 2017*), with population trajectories of some species indicating that

extirpation (*Jones et al., 2021*) or extinction (*Carpenter et al., 2008*) is likely. With extinction already expected to be imminent for some corals (*Sheppard, Sheppard & Fenner, 2020*; *Neely et al., 2021*), studies of the mechanisms by which corals might avoid this outcome are urgently required (*e.g.*, *Smith et al., 2014*). Recent work has highlighted the potential for various mechanisms to promote persistence of symbiotic corals, for example, through epigenetic effects, genetic adaptation, and variation in the complement of their viral, microbial, and algal symbionts (*Putnam, 2021*; *Lachs et al., 2023*). Further analyses of these effects are required to understand how corals will respond to changing environments, and to formulate conservation strategies that can best increase the ability to protect coral reefs (*Hoegh-Guldberg et al., 2018*; *Bellwood et al., 2019*).

Corals can respond to changing conditions through phenotypic plasticity (*i.e.*, acclimatization; *Putnam, 2021*), genetic adaptation by the host or their symbionts (*Matz et al., 2018*), or changes in symbiotic partners (*Cunning, Silverstein & Baker, 2015a*) to create emergent holobiont phenotypes. A total of 30 years ago, *Buddemeier & Fautin (1993)* proposed the adaptive bleaching hypothesis (ABH) which posited that shifts in algal symbiont taxa in corals could be beneficial in responding to elevated temperature. Much has changed since this contribution, notably to reveal genetic diversity within the Symbiodiniaceae (*LaJeunesse et al., 2018*), and to reinterpret the concept of coral "symbiosis" to embrace microbial, algal, and viral taxa (*van Oppen & Blackall, 2019*). It is now accepted that many (but not all) corals can undergo changes in holobiont composition (*van Oppen & Blackall, 2019*), particularly with respect to Symbiodiniaceae (*Cunning, Silverstein & Baker, 2015a*). In considering how corals will respond to changing conditions (*e.g.*, *Hughes et al., 2017*), and whether these responses could be harnessed in conservation efforts (*Voolstra et al., 2021*), expansion of ecological time series of coral communities (*e.g.*, *Howells, Bay & Bay, 2022*) to include genetic characterization of Symbiodiniaceae communities (*e.g.*, *Baker, 2003*; *Palacio-Castro et al., 2023*) is desirable.

Here we discuss the implementation of this vision on Caribbean reefs that have been studied for decades, but in a location where implementation of this plan is challenged by low coral abundance, high coral mortality, and a stringent permitting framework. The present study describes a time-series established in 2017 to test for variation in Symbiodiniaceae in scleractinian corals on the fringing reefs of St. John, U.S. Virgin Islands. The objective was to sample tagged colonies over multiple years to test for differences among species in the way their associated symbionts change over time. Corals were tagged and sampled in July and August 2017, 1 month before the island was hit with two Category 5 hurricanes (*Edmunds, 2019*). A second sampling took place in 2018 in a field season expurgated by the damage inflicted by the storms, and a third sampling was completed in the summer of 2019. We use our results to show how abundances of Symbiodiniaceae changed, and offer interpretations of these effects developed through the prism of conservation needs for coral reef ecosystems.

**Table 1 Summary of coral species tagged and sampled (number of colonies) at the three sites in each year.**

| Species | Cabritte Horn | | | East Tektite | | | White Point | | |
|---|---|---|---|---|---|---|---|---|---|
| | 2017 | 2018 | 2019 | 2017 | 2018 | 2019 | 2017 | 2018 | 2019 |
| *Orbicella annularis* | 2 | 1 | 1 | 1 | 0 | 0 | 2 | n/a | 2 |
| *Montastraea cavernosa* | 3 | 3 | 3 | 3 | 3 | 3 | 4 | n/a | 3 |
| *Orbicella faveolata* | 2 | 2 | 1 | 1 | 0 | 0 | 2 | n/a | 2 |
| *Orbicella franksi* | 1 | 1 | 1 | 1 | 1 | 2 | 2 | n/a | 2 |
| *Porites furcata* | 2 | 0 | 0 | 6 | 0 | 1 | 3 | n/a | 0 |
| *Diploria labrynthiformis* | 3 | 2 | 2 | 5 | 3 | 4 | 2 | n/a | 2 |
| *Colpophyllia natans* | 3 | 3 | 3 | 5 | 4 | 4 | 2 | n/a | 1 |
| *Siderastrea siderea* | 3 | 2 | 3 | 2 | 2 | 2 | 6 | n/a | 5 |
| *Pseudodiploria strigosa* | 2 | 1 | 2 | 3 | 3 | 2 | 4 | n/a | 3 |
| Total | 21 | 15 | 16 | 27 | 16 | 18 | 27 | n/a | 20 |

**Note:**
White Point was not sampled in 2018 (n/a).

# MATERIALS AND METHODS

## Project design and field sampling

In the summer of 2017, nine scleractinian species (Table 1) from six genera were selected based on their high abundance and ecological importance. A total of 75 coral colonies were sampled from 7–10 m depth at White Point, East Tektite, and Cabritte Horn on the south shore of St. John (Fig. S1), where the reefs have been monitored since 1992 (*Edmunds, 2018*). Colonies were haphazardly selected along 40 m transects at each monitoring site (*Edmunds, 2018*) between 27 July and 7 August 2017, and their positions were identified using cartesian coordinates. The colonies were selected based on ease of location and size (>20 cm diameter to support multiple samplings), and marked with aluminum tags epoxied to the substratum. Following tagging, a tissue biopsy (<1-cm diameter) was removed from the upper surface of each colony using stainless steel clippers. Samples were placed in Whirl-Pak bags and processed within 2 h of sampling.

We searched for the tagged colonies in 2018 (13–15 August), 11 months after Hurricanes Irma and Maria, and in 2019 (24–29 July). Colonies were located using underwater maps, site photographs, and an underwater metal detector (Vibraprobe 580, Treasure Products, CA) to find their tags. When located, a tissue biopsy was collected using the identical procedure established in 2017. Sampling in 2018 was limited due to the lingering effects of the hurricanes on island infrastructure. Sampling was comprehensive in 2019, and the reefs were exhaustively searched for tagged colonies to increase confidence that missing colonies were dead rather than not located. These collections were completed under permits issued by the Virgin Islands National Park (VIIS-2017-SCI-0037, VIIS-2018-SCI-0019, VIIS-2019-SCI-0023) and the Government of the Virgin Islands (DFW18088J, DFW19044J).

## DNA extraction and sequencing

On shore, coral biopsies were individually transferred to microcentrifuge tubes filled with 500 μL of 1% sodium dodecyl sulfate DNA buffer using sterilized forceps, and heated to 65 °C for 1.5 h. Samples were shipped to the Hawaiʻi Institute of Marine Biology where genomic DNA was extracted using an organic extraction protocol (*Baker & Cunning, 2016*). Symbiodiniaceae diversity was characterized through the amplification and sequencing of the ribosomal Internal Transcribed Spacer 2 (ITS2) region (*LaJeunesse, 2001*; *Davies et al., 2022*). DNA samples were sent to the University of Texas at Austin for library preparation with 'itsD' and 'its2rev2' primers (*Stat et al., 2009*) and sequencing on an Illumina MiSeq platform with 2 × 300 paired-end read chemistry. The paired forward and reverse reads from each sample were submitted to SymPortal to resolve ITS2 type profiles based on repeated co-occurrence of purported intragenomic variants (*Hume et al., 2019*).

## Data analysis

Symbiodiniaceae ITS2 sequence variants and type profiles within each sample were visualized as stacked bar graphs. Variation over time and among host species was visualized for each algal genus using non metric multidimensional scaling (NMDS) of Bray-Curtis dissimilarity matrices from relative (or square-root transformed) abundances. Differences among host species were tested by PERMANOVA, and rain cloud plots (*Allen et al., 2021*) were used to compare variation in symbiodiniaceaen communities between and within colonies by species. All data analysis was conducted using R (v4.0.0) (*R Core Team, 2020*).

# RESULTS

## Overview

In 2017, 75 coral colonies from nine species were tagged and sampled (Table 1). A total of 11 months after Hurricanes Irma and Maria when the 2018 sampling occurred, the benthos was disorganized with abundant coral rubble, and impairment of island infrastructure restricted the capacity to conduct fieldwork. As a result, some corals were not found as they had been dislodged and killed (*e.g.*, *Porites furcata*), limited underwater time restricted the capacity to find other colonies, and there was insufficient time to search at White Point. In 2018, 71% of the tagged colonies were found at Cabritte Horn, 59% at East Tektite, and none at White Point (Table 1). In 2019, more time was available for surveys, and 72% of all the tagged colonies were found and sampled (Table 1). We assume the remaining 28% had died.

## Symbiodiniaceae communities

ITS2 sequencing and processing by SymPortal (*Hume et al., 2019*) resulted in a dataset with 3,104,278 post-MED sequences across 160 samples. Ten samples with <1,000 sequences were removed from downstream analyses, and the remaining samples had a mean of 18,962 (±9,136 S.D.) sequences per sample (excluding one high outlier). Since

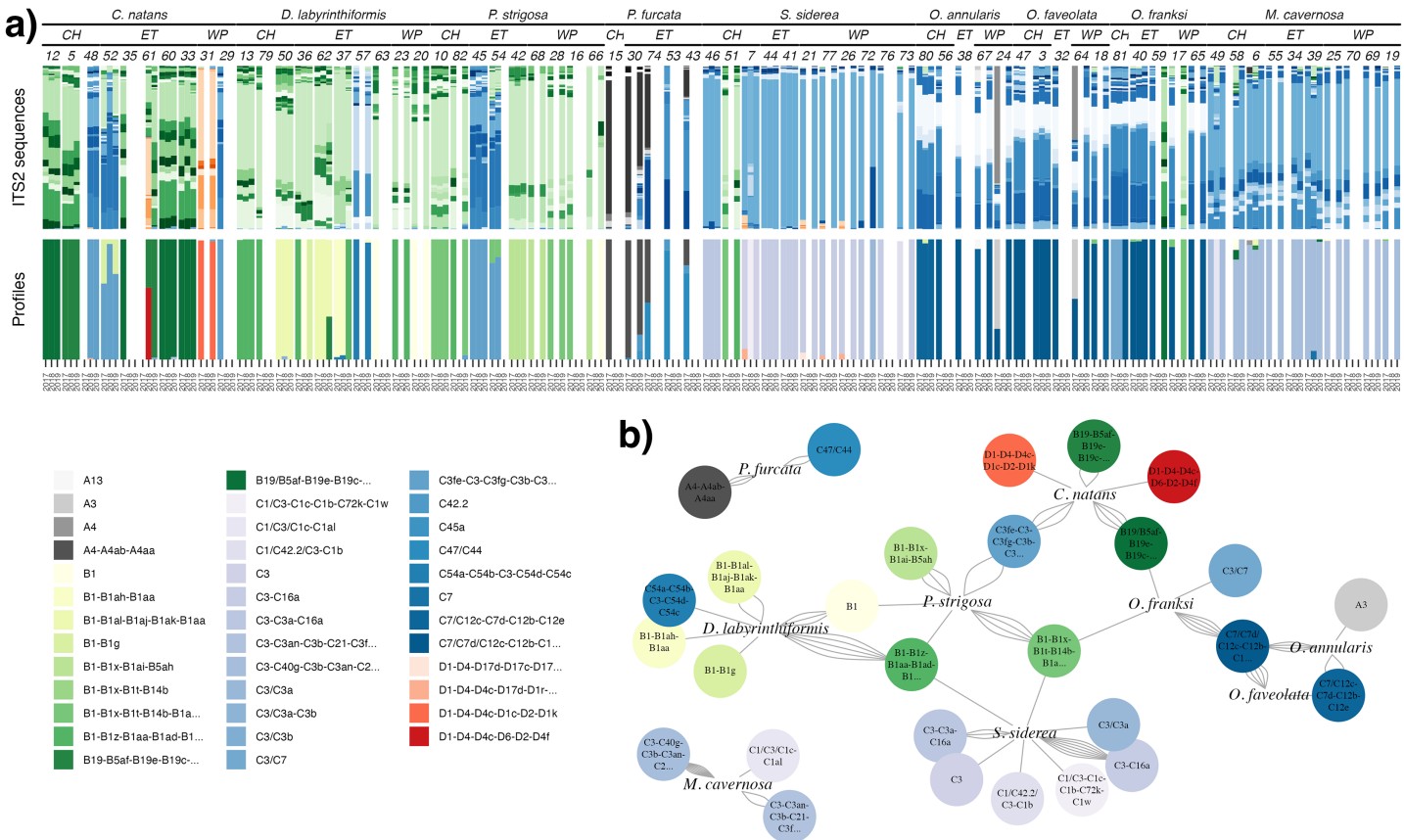

**Figure 1 Symbiodiniaceae communities.** (A) Relative abundance of all ITS2 sequence variants (>0.1%; upper bars) in each sample, with unique sequence variants given distinct colors (greys = *Symbiodinium*, greens = *Breviolum*, blues = *Cladocopium*, reds = *Durusdinium*). Samples (columns) are ordered by species, site (CH = Cabritte Horn, ET = East Tektite, WP = White Point), colony number, then year. The lower set of bars indicates the corresponding SymPortal ITS2 type profiles in each sample, with profile names and colors provided in the legend. (B) Network analysis of the associations between coral host species and dominant ITS2 type profiles, where the number of connections (network edges, gray curved lines) between each host and symbiont profile indicate the number of individual colonies of that coral species that were found dominated (>50% abundance) by that ITS2 type profile at any time.

Symbiodiniaceae are known for high intragenomic ITS2 sequence variation (*Davies et al., 2022*), SymPortal uses repeated co-occurrences of variants to collapse assumed intragenomic variation into putative taxa, or ITS2 type profiles (*Hume et al., 2019*). Here, we present and analyze results at the level of both sequence variants (Fig. 1A) and profiles (Fig. 1B) for complementary interpretation. Overall, sequences comprised 525 ITS2 sequence variants and 42 ITS2 type profiles belonging to *Symbiodinium* (27 sequences, four profiles), *Breviolum* (224 sequences, 13 profiles), *Cladocopium* (234 sequences, 21 profiles), *Durusdinium* (38 sequences, four profiles), and *Gerakladium* (two sequences, 0 profiles) (Fig. 1A).

Coral species hosted different Symbiodiniaceae communities (Figs. 1A and 1B). Faviids (*Colpophyllia natans, Diploria labyrinthiformis, Pseudodiploria strigosa*) were mostly dominated by *Breviolum*, and less commonly by *Cladocopium*, and two colonies of *C. natans* were the only corals found dominated by *Durusdinium*. Merulinids (*Orbicella annularis, O. faveolata,* and *O. franksi*) and *Montastraea cavernosa* were mostly dominated

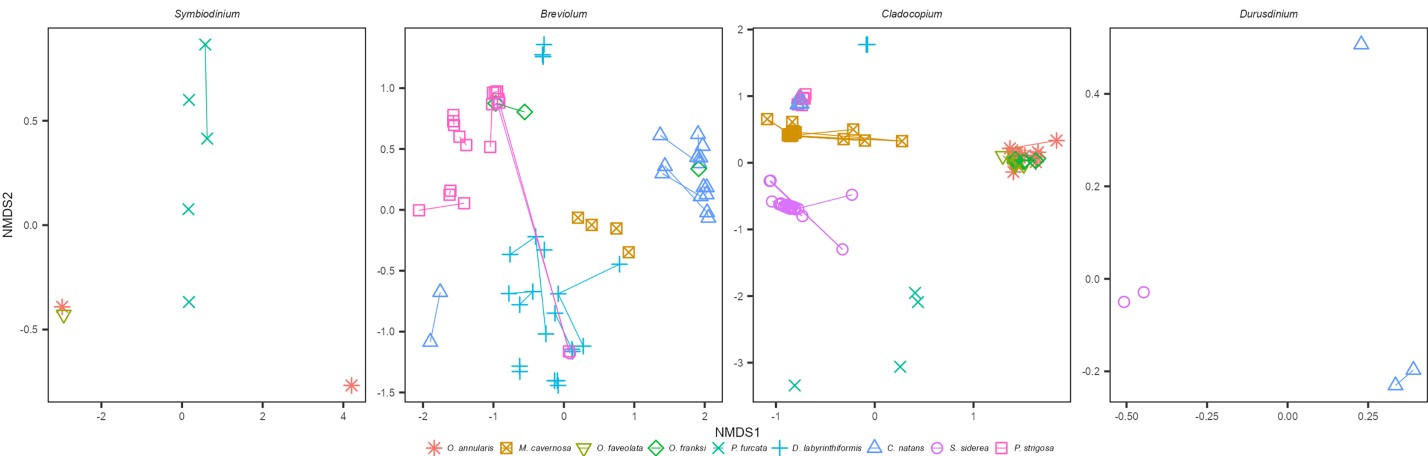

**Figure 2** **NMDS ordination of sequences within each Symbiodiniaceae genus, for samples where the genus was present at ≥5% relative abundance.** Points represent sequences within each Symbiodiniaceae genus from individual colonies, colored by host species, and samples from the same colony (different years) are connected by line segments.

by *Cladocopium*, and rarely by *Symbiodinium* or *Breviolum. Siderastrea siderea* (Rhizangiidae) was mostly dominated by *Cladocopium* (and rarely *Breviolum*), and *Porites furcata* (Poritidae) was associated with *Symbiodinium* and *Cladocopium* in varying proportions.

Within each genus of Symbiodiniaceae, distinct ITS2 type profiles were associated strongly, but not completely, with different coral species (Figs. 1A and 1B). The three Faviids, *C. natans, D. labyrinthiformis*, and *P. strigosa*, as well as *M. cavernosa*, tended to be dominated by species-specific *Breviolum* profiles (Fig. 1B). In contrast, when *O. franksi* (*n* = 2) and *S. siderea* (*n* = 1) colonies were occasionally dominated by *Breviolum*, these were profiles more commonly found in other host species, such as B1-B1x-B1t-B14b-B1af-B1ae-B1ag from *P. strigosa*, B19/B5af-B19e-B19c-B5ag-B19d-B19f-B40a-B19h from *C. natans*, and B1-B1z-B1aa-B1ad-B1ac-B1ab from *D. labyrinthiformis* (Fig. 1B). These patterns in host-*Breviolum* specificity were also supported by significant differences in composition of all *Breviolum* sequence variants in all pairwise comparisons of host species (PERMANOVA, *p* < 0.05), except those involving *S. siderea* or *O. franksi* (Fig. 2, Table S1, Fig. S2).

*Cladocopium* ITS2 profiles were also structured by host species (Figs. 1A and 1B); profiles characterized by C7 variants were mostly associated with *Orbicella* spp., C40g and C3an with *M. cavernosa*, C3fe with *C. natans* and *P. strigosa*, C54a and C54b with *D. labyrinthiformis*, C3a and C16a with *S. siderea*, and C47, C42.2, and C45a with *P. furcata. Orbicella* spp. were mostly dominated by the same *Cladocopium* profile (C7/C7d/C12c-C12b-C12e-C7h-C3-C7g; Fig. 1B), or very closely related ones (C7/C12c-C7d-C12b-C12e; C3/C7), and the composition of *Cladocopium* sequences was not different among *Orbicella* species (Fig. 2; PERMANOVA, *p* = 1). When *P. strigosa* and *C. natans* colonies were occasionally *Cladocopium*-dominated, it was by the same profile (C3fe-C3-C3fg-C3b-C3ag-C21; Fig. 1B) with similar sequence variant composition (Fig. 2; PERMANOVA, *p* = 0.87). Dominant *Cladocopium* profiles differed between all other host

species (Fig. 1B), as did the composition of *Cladocopium* sequence variants (Fig. 2; PERMANOVA, $p < 0.05$; except between *D. labyrinthiformis* ($n = 2$) *versus P. furcata* ($n = 4$) where low sample size restricted statistical power (Table S2, Fig. S3). The *Symbiodinium* and *Durusdinium* profiles also showed host species-specific associations (Figs. 1b and 2), although low sample sizes also reduced statistical power for these comparisons (PERMANOVA, $p > 0.1$).

Within colonies, changes in the dominant symbiont profile over time were uncommon. A change in the genus of the dominant symbiont occurred in one colony of *C. natans* (from *Durusdinium* to *Breviolum*), one colony of *O. annularis* (from *Symbiodinium* to *Cladocopium*), and two colonies of *O. franksi* (from *Cladocopium* to *Breviolum*) (Fig. 1A). A change in the dominant ITS2 type profile (within the same symbiont genus) occurred in another 13 coral colonies (representing all species except *P. furcata*, for which only one colony was repeatedly sampled). The remaining 70% of colonies sampled in multiple years ($n = 39/56$) had the same dominant ITS2 type profile at each time.

Changes in the abundance of non-dominant symbiont genera (representing <50% of sequences) within colonies over time were more common. Of all colonies sampled, 91% ($n = 64$) had at least one additional non-dominant genus at >0.1% relative abundance, and 47% ($n = 33$) had a non-dominant genus at >1% relative abundance. Among colonies sampled in multiple years, 48% ($n = 27$, including all species except *P. furcata*) showed a change in the number of genera at >1% relative abundance. However, changes over time in ITS2 assemblages were generally small and inconsistent, as revealed by small and irregular movement in 2-D ordination space (Fig. 3). The only consistent change over time was observed in *M. cavernosa*, where colonies showed an increase in non-dominant *Breviolum* from <1% in 2017 to ~10% relative abundance in 2018, followed by a reversion to <1% in 2019 (Figs. 1A and 3). These relative abundances should be interpreted with caution, as compositional change in mixed communities is complicated by unquantified yet potentially high copy number variation across taxa (*e.g.*, 1% of sequences may not reflect 1% of symbiont cells; *Davies et al., 2022*).

The generally low variability in Symbiodiniaceae communities over time within colonies contrasted with high variability observed among colonies (Fig. 4). Variation among colonies at any one time was greater (from 1.8-fold in *O. annularis* to 9.5-fold in *C. natans*) than variation within a colony over time for all species except *M. cavernosa* (and *P. furcata* for which only one colony was re-sampled).

## DISCUSSION

Scleractinian corals are drastically declining in abundance on many coral reefs (*Jackson et al., 2014*; *Hughes et al., 2017*), and there is an urgent need to understand the implications of rapidly dwindling coral populations. Yet despite this urgency, the extent to which one of the most important aspects of reef corals, their Symbiodiniaceae communities, changes over time is not fully understood (*Thornhill et al., 2006*; *Edmunds et al., 2014*; *Cunning, 2021*; *Palacio-Castro et al., 2023*). Many corals harbor multiple Symbiodiniaceae taxa (*Silverstein, Correa & Baker, 2012*), often with numerous rare genotypes (*Quigley et al., 2014*), and both the types and relative abundances of these algae can change in some corals

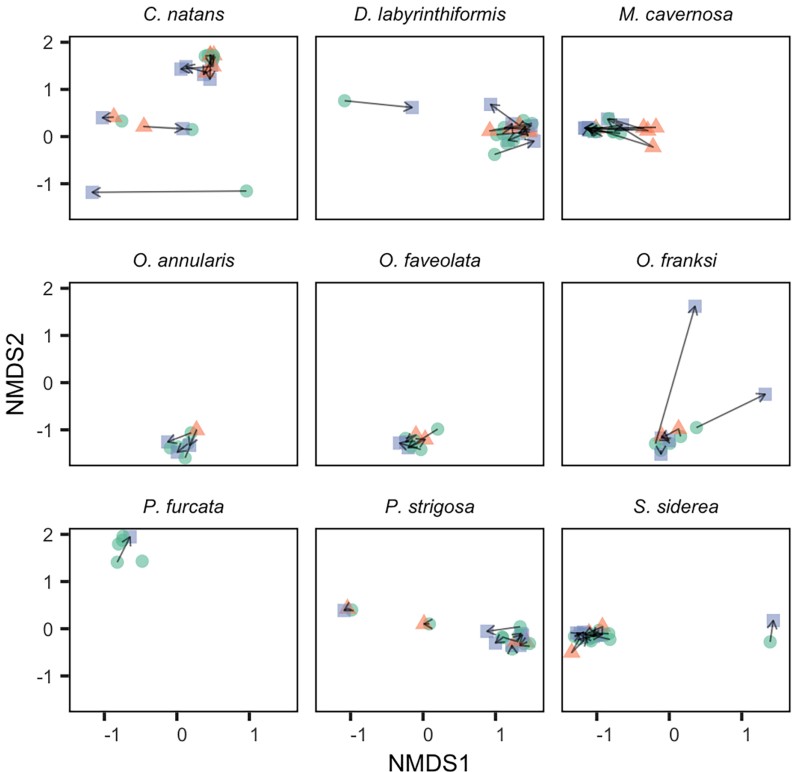

**Figure 3 Change in Symbiodiniaceae community structure over time within individual tagged colonies.** NMDS was performed on square-root transformed relative abundances of all ITS2 sequence variants. Samples collected in 2017 (green circles), 2018 (red triangles), and 2019 (blue squares) from the same tagged individual are connected by black arrows.

over time (*Chen et al., 2005*; *Cunning, Silverstein & Baker, 2018*; *Palacio-Castro et al., 2023*), which can better match the holobiont phenotype to environmental conditions (*Buddemeier & Fautin, 1993*). Beneficial changes in symbiont types may therefore increase holobiont fitness (*Buddemeier et al., 2004*) in response to environmental fluctuation or disturbance, such as by reducing the sensitivity of corals to thermal stress (*Silverstein, Cunning & Baker, 2015*). Motivated by these discoveries, the present study was initiated to augment ecological analyses of coral reefs in St. John (*e.g.*, *Edmunds, 2018*, *2019*) with the ability to describe how their algal symbionts are changing at the same time.

Our results contribute to the fields of coral biology and conservation biology charged with managing coral reefs to enhance their ecological persistence (*Hoegh-Guldberg et al., 2018*; *Bellwood et al., 2019*). First, we reveal a high diversity of Symbiodiniaceae in nine species of corals, and demonstrate high variation among host species and colonies. In all species sampled, the dominant symbiont type profile varied among colonies, and many of these profiles were relatively rare across all colonies. Second, the symbionts within colonies were relatively stable over 3 years, despite substantial impacts from physical (*i.e.*, hurricanes (*Edmunds, 2019*)) and biological disturbances, for example, stony coral tissue loss disease (SCTLD), which is thought to infect Symbiodiniaceae algae rather than the host (*Beavers et al., 2023*). This disease was first reported in the US Virgin Islands (St.

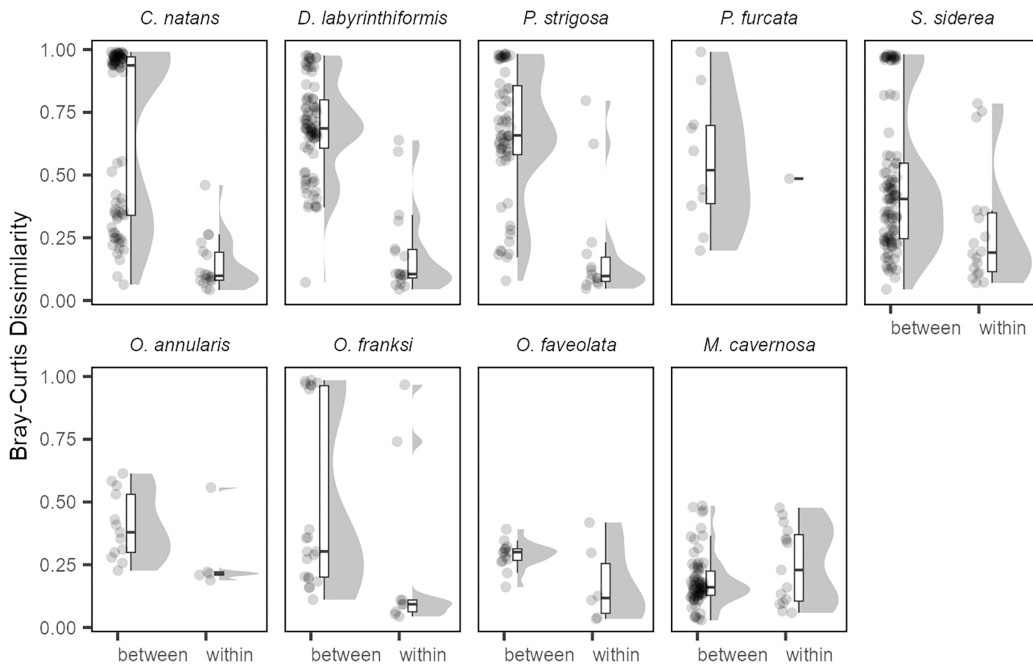

**Figure 4** **Differences in Symbiodiniaceae communities between colonies (sampled in the same year) and within the same colonies (sampled in different years) for each coral species.** All pairwise Bray-Curtis dissimilarities by group are visualized as rain cloud plots with boxplots, scatter plots of data (left for each category), and probability density function distributions (right for each category). Box plots show median and interquartile ranges (IQR) with whiskers showing the range of values within 1.5 * IQR.

Thomas) in January 2019 (*Brandt et al. 2021*), and although it is likely that it was affecting corals in St. John (~3.5 km east of St. Thomas) throughout 2019, mortality from this cause was not prominent on the south shore of St. John until the summer of 2020 (PJ Edmunds, 2020, personal observation). These two major findings from our study support the inference that algal diversity within coral host species may be lost as coral abundances decline, especially if colonies with select symbiont taxa are more vulnerable to disturbances (*Starko et al., 2023*). Wide-scale mortality of coral colonies could therefore lead to reef-scale depletion of Symbiodiniaceae diversity, unless these symbionts are maintained at low abundances within colonies dominated by other symbiont taxa, or retained in other microhabitats within the reef ecosystem (*Cunning et al., 2015b*; *Fujise et al., 2021*). Reduced Symbiodiniaceae diversity or availability could also constrain the potential for future beneficial changes in symbiotic algal communities within corals (*sensu Quigley, Bay & Willis, 2017*).

Given previous observations of changes in algal symbionts in Caribbean corals over time (*Baker, 2003*; *Cunning, Silverstein & Baker, 2015a*), when the present study began in 2017, it was reasonable to expect that some changes would be detected at either the genus or type profile level. This expectation was reinforced by evidence that the abundances of different symbiont genera within *Orbicella annularis* at 14-m depth in Lameshur Bay changed between 1994 and 2010 in at least some colonies (*Edmunds et al., 2014*). Moreover, the two Category 5 hurricanes followed by heavy rains and turbid seawater in

the autumn following initial sampling in 2017 (*Edmunds, 2019*) represented large disturbances to which symbiodiniacean communities in corals might be expected to respond (cf *Kennedy et al., 2016*; *Claar et al., 2020*). While our results reveal a high variation among species and colonies (Fig. 1), consistent with contemporary expectations (*Baker, 2003*; *Dougan et al., 2022*), the change within colonies over 3 years was small.

While it was beyond the scope of this study to address the biological meaning of this temporal stability, there are three hypotheses that are consistent with the present results. First, it is possible that the disturbances occurring in St. John over 2017–2019 were not effective in eliciting a change in Symbiodiniaceae communities. Second, by sampling at ~12 month intervals, it is possible that changes and reversions went undetected. Third, it is possible that the Symbiodiniaceae were challenged by ecologically meaningful disturbances, and that their stability reflects high resistance to the disturbance regimes. The value of testing these hypotheses in the future provides compelling reasons to further develop symbiodiniacean time series for Caribbean corals, but as we describe below, there are challenges in realizing this vision.

In the Caribbean, corals are rapidly succumbing to mortality on a scale of years (*Jackson et al., 2014*; *Edmunds, 2018*, *2019*), making repeated sampling of tagged colonies a risky prospect, because tagged corals are likely to die before being sampled more than once. The high probability of this outcome (*i.e.*, dying before subsequent sampling), along with the general stability in symbiont community structure that we observed within colonies over time, suggests that sampling to develop a time series for Symbiodiniaceae communities in corals on coral-depleted Caribbean reefs should target a large number of randomly selected corals rather than a small number of corals that are permanently marked. Such random sampling could maximize the number of different colonies sampled, which is where most variation in algal taxa was detected here. Moreover, this sampling design would capture changes in the relative abundance of coral colonies with different symbiont taxa that may occur through differential mortality, providing insights into how disturbances and stressors might select for certain host-symbiont combinations. Indeed, random sampling may better reveal ecologically meaningful changes in Symbiodiniaceae communities by capturing both symbiont shuffling and differential mortality of host-symbiont associations, even though it cannot distinguish between these two mechanisms. When the mechanism causing Symbiodiniaceae communities to change is of particular interest, then tagged corals should be resampled, or a hybrid approach considered, in which some colonies are tagged for repeated sampling and some colonies are randomly selected (and not tagged). However, when expecting that symbiont shuffling may be infrequent, that differences in Symbiodiniaceae communities between colonies may be high, and that there are elevated risks of individual colonies dying, then random sampling may be a more effective and viable strategy to monitor Symbiodiniaceae changes on reefs.

Our study has implications for conservation biology as it pertains to coral reefs, because the degraded state of many reefs (*Hughes et al., 2017*), and the likely future trajectories of declining coral populations (*Riegl, Berumen & Bruckner, 2013*; *Mason, Bozec & Mumby, 2023*), argues for solution-oriented science that can aid in conservation. Part of such efforts

requires augmentation of ecological monitoring to support tests of the genetic basis of survival and mortality in coral hosts and their symbionts (*Howells, Bay & Bay, 2022*). Yet at a time when the needs for this work are acute, the permitting frameworks to support such efforts are becoming more conservative as the natural resources being protected dwindle in size. The logistical requirements for this type of work typically include an extensive application process for permits to collect biopsies, particularly for less common species. Permission also must be sought to permanently mark coral colonies, and in many cases, international permits must be secured to export DNA for analysis.

As we describe here for St. John, corals that are attractive targets for genetic time series of their symbionts have a high chance of dying before multiple samplings occur, and if they do not die, their low abundance requires extensive searching to relocate them, and the tagging materials require regular maintenance. Our results suggest that random sampling of large numbers of corals may be more tractable with higher returns on the investment of time and resources, notably with sampling to support species contrasts of algal symbiont types (and relative abundances) at each timepoint. Such an approach would generate higher community-level resolution of symbiont dynamics over time compared to repeatedly sampling tagged colonies.

## ACKNOWLEDGEMENTS

We dedicate this manuscript to Ruth D. Gates, who played an integral role in the conception and design of this study, and whose memory inspires us to better understand and protect coral reefs. Comments from N Kriefall, K Hoadley, and one anonymous reviewer improved an earlier draft of this article. This is contribution number 385 of the CSUN Marine Biology Program.

### Funding

This research was funded by the National Science Foundation (to Peter Edmunds, DEB 13-50146). Additional financial support was provided to Elizabeth Lenz by a NOAA John A. Knauss Marine Policy Fellowship, and to Ross Cunning by NSF OCE-1851305.
The funders had no role in study design, data collection and analysis, decision to publish, or preparation of the manuscript.

### Grant Disclosures

The following grant information was disclosed by the authors:
National Science Foundation: DEB 13-50146.
NOAA.
Knauss Marine Policy Fellowship.
National Science Foundation: OCE-1851305.

### Competing Interests

The authors declare that they have no competing interests.

## Author Contributions

- Ross Cunning conceived and designed the experiments, performed the experiments, analyzed the data, prepared figures and/or tables, authored or reviewed drafts of the article, and approved the final draft.
- Elizabeth A. Lenz performed the experiments, analyzed the data, authored or reviewed drafts of the article, and approved the final draft.
- Peter J. Edmunds conceived and designed the experiments, performed the experiments, authored or reviewed drafts of the article, and approved the final draft.

## Field Study Permissions

The following information was supplied relating to field study approvals (*i.e.*, approving body and any reference numbers):

This research was completed under permits issued by the Virgin Islands National Park (VIIS-2017-SCI-0037, VIIS-2018-SCI-0019, VIIS-2019-SCI-0023) and the Government of the Virgin Islands (DFW18088J, DFW19044J).

## DNA Deposition

The following information was supplied regarding the deposition of DNA sequences:

The ITS2 sequence datas are available at NCBI: PRJNA1106054.

## Data Availability

All analysis code is available at GitHub and Zenodo:

- Ross Cunning, & Beth Lenz. (2024). jrcunning/STJ_ITS2: PeerJ publication release (v1.0). Zenodo. https://doi.org/10.5281/zenodo.10966764.

## Supplemental Information

Supplemental information for this article can be found online at http://dx.doi.org/10.7717/peerj.17358#supplemental-information.

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
