# Peer review of "Measuring multi-year changes in the Symbiodiniaceae algae in Caribbean corals on coral-depleted reefs"

_PeerJ, doi:10.7717/peerj.17358_

## Round 0.1 · original submission · Minor Revisions

Three experts in this field have evaluated your manuscript and all have high praise for your contribution, as do I. All reviewers have suggested some minor revisions which need to be attended to in a revised version of the manuscript. Please ensure that you answer each comment in a rebuttal and upload the letter along with a tracked changes version and a clean version of the manuscript. Congratulations on a great contribution.

·

Basic reporting

All was well except the quality of the figures could be improved, they look very grainy in the pdf I was provided. My understanding wasn't impeded by this in Figures 3 and 4, but it definitely was for Figures 1 and 2.

Experimental design

No comment, all is well.

Validity of the findings

No comment, all is well.

Additional comments

Despite presenting a huge dataset, the paper was very succinct. I commend the authors on a thorough investigation and presentation. I have some comments for improving clarity throughout the manuscript and then one comment on substance at the end of the discussion:

Abstract
L42-44: This sentence (beginning with ‘Seventy-five corals…’ and ending with ‘…have died’) was very wordy and hard for me to understand during my first read-through with the percentages. A suggestion: ‘Seventy-five corals from 9 species were marked and sampled in 2017. Of these colonies, 41% were non-exhaustively sampled in 2018 and 72% were exhaustively sampled in 2019; 28% could not be found and were assumed to have died.’ Or something of the like. I took out the mention of the hurricanes here because you mention it again in the next sentence so that felt repetitive.
L47-48: I suggest adding in (2017) after the mention of the hurricanes, and adding the year of when SCTLD arrived in St. John for context.
L47-48: I think SCTLD spelled out has capital letters at the beginning of the words – I suggest changing to Stony Coral Tissue Loss Disease.
L50: I suggest changing to “Together,” with comma

Introduction
L67-8: can ‘changes in the host genotype’ be changed to ‘adaptation of the host genome’ if that’s what was meant? I’m not sure what a change in the host genotype is implying otherwise
L73-76: The sentence beginning with “Corals can respond to changing…’ and ending with ‘…emergent holobiont phenotypes’ – I felt all of this is essentially repeated from lines 66-69. I would recommend cutting it.
L87-90: this last sentence of the paragraph (beginning with ‘Here we discuss…’) fits better with the next paragraph. It confused me to say “location” here without saying what location it is, but you clear that up in the first sentence of the next paragraph, so I would put the two sentences together if possible.
L92: add ‘Virgin Islands’ information after St. John?
L93-94: I didn’t feel that “the way that they change over time” was clear – can you change to something like “to test for differences among coral species in the way their associated symbionts change over time.”
L94-95: When it says “Corals were tagged in July and August 2017, and sampled one month before the island was hit with two Category 5 hurricanes” – I’m not sure whether you’re saying you sampled at a later point than when the tagging occurred. Can you put something like “Corals were tagged in July and August 2017, approximately one month before the island was hit…” if that retains the intended meaning.

Materials & Methods
L103-110: I think it’s better to remove the hyphens in the units for things like ’40-m transect’ and ’>20-cm diameter’, becoming ’40 m transect’ and ‘>20 cm diameter’, respectively, but I could be wrong.
L105: I would repeat St. John, US Virgin Islands information after the names of the reefs
L108: I would replace ‘They were selected based…’ at the beginning of the sentence with ‘The colonies were selected based…’
L109: I would remove the comma before ‘and size’ to become ‘…on ease of location and size…’
L114-116: The list was confusing as worded, what about “Colonies were located using underwater maps, site photographs, and an underwater metal detector (Vibraprobe 580, Treasure Products, CA) to find their tags.
L144: remove hyphens after ‘between-‘ and ‘within-‘, I think

Results
L198: I would remove the spaces within 'p = 0.87', they’re not present around all the other p-values in this section
L200: All the other p-values say ‘PERMANOVA’, so I think that’s missing before the p<0.05 in this line
L203: All the other non-significant p-values are given, for instance p=0.87 or p=1 on previous lines, so I would change p>0.1 to the actual value.
L212 onwards: Is this paragraph still referring to type profiles? Just want to make sure it’s clear, or whether it’s shifted to ITS2 sequences

Discussion
L251: Capitalize the words making up SCTLD, I think
L263: Can you define more what you mean by ‘expected changes’ – at the genus level? Type profile level?
L264: Changed how, changed in dominant genus (formerly clade) or?
L278: I’m not sure ‘resilience’ (recovery back to baseline levels after a disturbance) is the right word here in hypothesis number 3, because then this hypothesis is essentially the same as #2 where changes went undetected because they went back to how they were before which would indicate high resilience. Does it keep your intended meaning to say “their stability reflects high resistance to the disturbance regimes” (I would also change the ‘disturbances regimes’ as currently worded to ‘disturbance regimes’.)
L282-316: All three of these final paragraphs felt like it was making the same point over and over with very similar wording in each, saying that sampling a few colonies thoroughly might not be better than larger survey sampling at random. I would recommend heavily reorganizing or rewording these paragraphs to better distinguish them from each other, or simply cutting things that were repeated.
In addition, as troublesome as it is, I still think there’s value in sampling the same colony over again. Most papers with larger time point surveys at random, like you suggest doing instead of the current design, always have to acknowledge that they don’t know whether any observed Symbiodiniaceae community shifts over time are due to previously sampled colonies dying out OR the original colonies are alive but slightly different colonies were sampled at subsequent time points at random OR the same colonies were sampled again that changed their communities. I think your paper adds evidence that it’s previous colonies dying out in this area since huge temporal changes in the same colonies weren’t noted, but then didn’t the Edmunds 2014 paper appear to differ? As such, I wouldn’t throw the practice entirely out of the window, especially if different results are found in other areas where sampling frequently is more logistically feasible. I would hope there was more room for the value of both sides of this in the discussion.

Figures & Tables
Figure 1 – too low quality for me to see ‘2017-2019’ on x-axis of stacked bar plots
Figure 4 – is this type profiles or sequences?

Reviewer 2 ·

Basic reporting

This study tracks the dinoflagellate symbiont composition of corals of St. John over three years to assess symbiont diversity within and between coral species and colonies. This is conducted in the context of the effects of climate change on reefs and the physiological impacts of diverse symbiont species/strains on coral health and survival. Multi-year field datasets such as these are highly valuable but are difficult to sustain, as many funders are unwilling to support “unsexy” monitoring projects. I applaud the authors for the effort and hope they can continue to monitor corals these sites. The paper is well-written, the methods are sound and well-established, and it is a solid contribution to the literature. Congratulations to the authors. I have no major concerns or comments, only a few points below.

The authors argue that tagging colonies for repeated sampling is a vulnerable design, as the mortality rate of colonies is high enough that a meaningful portion of study corals may be lost during the course of a typical experiment. I understand the logic, though I’m not entirely convinced. If coral mortality is sufficiently high that the majority of (randomly selected, independent) tagged colonies may not survive over the span of a few years, then the sampling method is rendered irrelevant as total mortality will rapidly approach 100% and that reef is already lost. This may not be true for a few very rapidly growing/maturing species that can reach reproductive age in only a few years, but my imperfect understanding is that all of these species (except perhaps P. furcata?) take much longer to reach reproductive size. There is information to be gained from repeatedly measuring colonies, but as the authors recognize, the choice of sampling method may largely be a pragmatic decision for each study based on logistical and regulatory restrictions.


Fig. 1: The bottom labels of panel A are illegible, but appear to be sampling years; consider another way of presenting this information (grayscale horizontal bars for each colony?)
Table 1. Consider presenting this as a figure, e.g. bar graph.

The references cited section needs considerable cleanup prior to publication.

Experimental design

Appropriate.

Validity of the findings

Sound and well-reasoned.

·

Basic reporting

This paper showcases genetic variability (of the photosymbionts) across Caribbean coral colonies through space and time. The intent of this paper is highly valuable and timely given identified priorities for conservation and restoration efforts throughout the Caribbean Sea and further abroad. While there are now numerous ways to characterize and quantify symbiont genetic diversity within a reef coral, the authors have appropriately chosen a technique that allows for a relatively robust analysis. However, like any other technique, ITS2 sequencing and the Symportal pipeline have a few caveats which need to be addressed to best interpret the results. My relatively minor concerns in this area are outlined below. Overall, I think this manuscript (with minor revisions) can be an excellent contribution to the PeerJ community.

Experimental design

Methodology/interpretation: The ITS2 region of coral photo symbionts suffers from two very common problems in the realm of algal metabarcoding.

The first is copy number, which can differ across symbiont genera (symbiodinium vs. cladocopium vs. durusdinium) which can make interpreting barplots potentially misleading (without some normalization). This issue and potential means for resolving it are discussed in the symbiont diversity review paper (Davies et al., 2023 – PeerJ). It is unlcear if any normalization was done to account for this across genera. However, given that the authors did see some major shifts (albeit infrequently) between symbiont genera, such normalization could play a critical role in the interpretation.
The second issue is multiple copy variants of ITS2 within the same alga clonal variant (see Thornhill et al., 2007 – Molecular Ecology). The first barplot in Figure 1 reflects sequence variants of ITS2 within each sample but how this is interpreted can lead to different conclusions. One interpretation could suggest a diverse assemblage of algal photosymbionts within the same coral colony. However, if we assume multiple copy variants, the sequence variants in the barplot may simply reflect a single clonal variant of the species/profile reflected in the second set of barplots in figure 1. It is unclear how this data is being interpreted and perhaps some discussion of the caveats and assumptions being applied in this paper could help the reader better understand and interpret the results and discussion.

Validity of the findings

Results: It is difficult to understand how the statistics are being used to describe variability in sequence identity across species. Specifically, there are a couple instances where insignificant p-values are used to support results and it is not clear how they support the finding/descriptions. Consider revising lines 189-203.

Discussion: I have no issues with the discussion as written but I do think the data (high photosymbiont variability across colonies) could also support some discussion on colony variability in response to environmental stress. Could photosymbiont genetic variability (and the presumed physiological variability that comes with it) play a role in how colonies respond to environmental stress? Different genera of photosymbionts certainly impact colony response, but less attention has been placed on how different clonal variants of the same symbiont species may also impact holobiont response. Given the conservation angle of the manuscript, discussion on this topic could be well received.

Additional comments

Nice study, and timely. Congrats.

Kind Regards,

Kenneth D. Hoadley

---

## Round 0.2 · accepted · Accept

I am satisfied with the changes made to the manuscript and recommend that this submission be accepted for publication.